# *Actinomyces* sp. Presence in the Bone Specimens of Patients with Osteonecrosis of the Jaw: The Histopathological Analysis and Clinical Implication

**DOI:** 10.3390/antibiotics11081067

**Published:** 2022-08-05

**Authors:** Norliwati Ibrahim, Nurul Inaas Mahamad Apandi, Syafiqah Aina Shuhardi, Roszalina Ramli

**Affiliations:** 1Department of Craniofacial Diagnostics & Biosciences, Faculty of Dentistry, Universiti Kebangsaan Malaysia, Kuala Lumpur 50300, Malaysia; 2Department of Oral & Maxillofacial Surgery, Hospital Canselor Tuanku Muhriz, Universiti Kebangsaan Malaysia Medical Centre, Kuala Lumpur 56000, Malaysia; 3Department of Oral & Maxillofacial Surgery, Faculty of Dentistry, Universiti Kebangsaan Malaysia, Kuala Lumpur 50300, Malaysia

**Keywords:** actinomycosis, *Actinomyces* sp., osteoradionecrosis, medication-related osteonecrosis of the jaw

## Abstract

Medication-related osteonecrosis of the jaw (MRONJ) and osteoradionecrosis (ORN) are two similar bone pathologies in the jaw with different aetiologies. Actinomycosis is a relatively rare oral infection caused by the Gram-positive anaerobe *Actinomyces* sp. that normally colonizes the oral cavity. Actinomycosis is associated with the pathogenesis of both the MRONJ and ORN, as evident in our cases, and not just as a superficial contaminant. The clinical and histopathological aspects of the cases treated in our centre were also reported with a review of the literature. Clinical implication on the treatment of the cases was highlighted in view of the presence of this microorganism.

## 1. Introduction

The medication related osteonecrosis of the jaw (MRONJ) and osteoradionecrosis (ORN) are the two most described osteonecrosis of the jaw (ONJ) [1]. These two diseases were reported to have compromised integrity of oral mucosa hence enabling microorganism entry into the intraoral tissue [2]. Consequently, there will be bone necrosis, sequestrum formation, fistula growth, and inflammatory infiltrates. In addition, actinomycosis attributes to the poor outcome of the ONJ [2] thus, the management of this infection is crucial. Actinomycosis was initially believed to be a superficial contaminant however, and later it was proven that the sulphur granules of the *Actinomyces* sp. were exclusively observed in areas within the bone necrosis [3].

Risk factors of ONJ with concomitant actinomycosis include patients with poor oral hygiene, a history of mucosal trauma post-dental extraction, male gender, diabetes, immunosuppression, and malnutrition. MRONJ patients were reported to be on antiresorptive agents like bisphosphonates or denosumab [4,5] or chemo-therapeutic agents and other immunosuppressive drugs [5], while the ORN patients were patients with a history of radiation therapy to the head and neck. To date, literature has described various antibiotic regimes for MRONJ as well as ORN patients. 

In this case series, we aim to describe the clinical as well as the histopathological characteristics of the sequestrum specimen from the MRONJ and ORN cases. The relevant antibiotic regime is proposed.

## 2. Methodology

A retrospective histopathological evaluation was performed on specimens selected from patients’ records from 2012 to 2021, previously subjected to biopsy of the jaw bones and diagnosed as either ORN or MRONJ. In addition, the patients’ records were reviewed based on these definitions: (i)MRONJ: (1) Current or previous treatment with antiresorptive therapy alone or in combination with immune modulators or antiangiogenic medications; (2) Exposed bone or bone that can be probed through an intraoral or extraoral fistula(e) in the maxillofacial region that has persisted for more than eight weeks; (3) No history of radiation therapy to the jaws or metastatic disease to the jaws. MRONJ has stage 0–3 based on patient’s symptoms, clinical and radiological findings [6].(ii)ORN: irradiated bone becomes devitalized and exposed through the overlying skin or mucosa without healing for three months, without recurrence of the tumour [7]. 

### The Specimen

The selected specimens were obtained from the existing paraffin-embedded blocks from the Oral Pathology Laboratory, Faculty of Dentistry, Universiti Kebangsaan Malaysia. The microscope slides were cut into 4-µm sections and stained with hematoxylin and eosin (H&E). The staining procedure of H&E was performed before the histological analysis. This procedure consisted of dewaxing and rehydration stages. During the dewaxing process, the slides were treated with xylene I, xylene II, and xylene, at five minute intervals for each solution. The slides were then dehydrated with absolute alcohol, 90% alcohol, and 80% alcohol for three minutes each. This was followed by washing the slides under running tap water for three minutes. The slides were then stained with hematoxylin for seven minutes and washed again for seven minutes under running water to remove excess stains. Following that, these slides were counterstained with eosin for five minutes. The slides were subjected to a dehydration procedure again by immersion in 70% alcohol, 80% alcohol, 90% alcohol, and finally in absolute alcohol. To finish, the slides were cleaned with xylene I, xylene II, and xylene for five minutes in each solution. The completed slides were mounted in the DPX mounting medium and covered with a coverslip. They were labelled accordingly and ready for histological analysis [8]. 

In the case series, *Actinomyces* sp. were histologically detected in all the eight cases through histopathology examination using the H&E staining. The examination revealed actinomycosis as basophilic masses with eosinophilic terminal clubs on staining with H&E [4]. Yellowish sulphur granules were formed by collection of bacteria trapped in the biofilm. Adjunct special stainings like Periodic Acid Schiff (PAS), Gömöri’s methenamine silver stain, fluorescein-conjugated specific antibodies, and Gram-stain may also be used to enhance the appearance of this microorganism. Gram positive filamentous microorganisms and the sulphur granules appearance strongly support the diagnosis of actinomycosis [1,4,9]. 

Clinically, all the eight patients presented with exposed bone in their oral cavities with or without the presence of pus. The cases with pus had culture and sensitivity tests and were inconclusive for *Actinomyces* sp. The presence of *Actinomyces* sp. were confirmed by the histopathological examination, as shown in Table 1.

In addition, comparison of the histopathological findings, i.e., necrotic bone, osteoclast, osteoblastic rimming, reactive bone, empty lacunae, inflammation, blood vessels, hyperaemia and thrombosis and presence of microorganism between the ORN and MRONJ specimens were performed. This examination was carried out by two oral pathologists in a simultaneous session.

## 3. Results

### 3.1. Brief Description of the Clinical Cases

Case 1: A 55-year-old Malay male was diagnosed with a malignant giant cell tumour (GCT) of his right knee with metastasis to his lungs. The malignancy was diagnosed in 2018. An orthopantomogram showed an area of radiolucency at the edentulous area of 35 to 41 (Figure 1A) while the computed tomography (CT) scan showed features of an osteolytic lesion admixed with sclerosis. Sequestrectomy of the left mandible area was performed. 

Case 2: A 64-year-old Malay man, with underlying multiple myeloma IgG Kappa, hypertension, and dyslipidemia. A biopsy of the necrotic bone involving his left maxillary sinus was performed. The radiograph for this case is as depicted in Figure 1B. The result was MRONJ with actinomycosis. Thus, he underwent surgical closure of the oro-antral communication of the left maxilla with augmentation of a free fat graft harvested from his abdomen. 

Case 3: A 72-year-old Chinese female had a history of nasopharyngeal carcinoma (NPC) and completed radiotherapy in 1994. She had a dental extraction of tooth 18 in 2008 and resulted with a non-healing socket (Figure 1D). An intraoral fistula on the right maxilla with exposed bone and signs of chronic infection was noted. A 1 cm × 1 cm length of bony exposure was observed at the right maxillary tuberosity area. 

Case 4: A 66-year-old Chinese female with a history of osteoporosis and systemic lupus erythematosus (SLE) was prescribed bisphosphonate tablets (Fosamax). She had tooth 35 extracted and subsequently developed exposed bone at the extraction socket with swelling and pus. Cone beam CT (CBCT) showed an area of sequestrum extending from tooth 35 to 32. 

Case 5: A 53-year-old Malay female with a known case of NPC completed radiotherapy and chemotherapy in 2006. There was a non-healing socket associated with the extracted tooth 48 with the presence of pus. The sequestrum was sent for a histopathological examination. 

Case 6: A 66-year-old Chinese male with a diagnosis of oral squamous cell carcinoma (OSCC) on his right lateral border of the tongue completed chemotherapy and radiotherapy in 2008. ORN of his bilateral mandible was diagnosed and sequestrectomy was performed in 2015. Figure 1C shows this patient’s bone condition. A year later, he underwent another sequestrectomy. 

Case 7: A 47-year-old Chinese male had his left lacrimal squamous cell carcinoma (SCC) surgically removed in 2013 with a total maxillectomy and left eye exenteration. He had a vertical rectus abdominis myocutaneous (VRAM) flap for reconstruction and then underwent radiotherapy. Consequently, he developed acute maxillary sinusitis secondary to right maxillary ORN. 

Case 8: A 69-year-old Chinese male with OSCC of the right lateral border of the tongue in 2010. He underwent a hemi-glossectomy and completed radiotherapy in the same year. Following a certain period of review, a sequestrum was noted in the area of mandible symphysis. The honeycomb, grey, necrotic bone was easily scraped off and separated from the basal bone. This patient had several series of surgery to remove the sequestrum as the infection recurred.

In total, there were five cases of ORN and three cases of MRONJ. Table 1 further describes the clinical and histological characteristics of the patients. Table 2 shows a comparison of the histopathological features between ORN and MRONJ. 

### 3.2. Histopathological Examination

Figure 2 below shows representative microscopic characteristics of the specimens. The findings were itemized in Table 2 to evaluate for a certain pattern. 

Based on the eight cases, the histological findings were almost similar for both ORN and MRONJ, except for osteoblastic rimming being absent in ORN and devoid of blood vessels in MRONJ.

### 3.3. Presence of Actinomyces sp.

Figure 3 below shows a bacterial colony of *Actinomyces* sp. in one of the cases.

## 4. Discussion

*Actinomyces* sp. is a normal commensal of the oral cavity and commonly found in the saliva, dental plaque, decayed teeth, gingival grooves, and tonsillar crypts. It is a relatively rare chronic infection with a low degree of virulence. However, under certain conditions that compromise the oral mucosal barriers and host susceptibility, the pathogenic *Actinomyces* sp. leads to actinomycosis infection. In addition, poor oral hygiene, non-healing extraction socket, and surgical manipulation may facilitate the entrance of *Actinomyces* deeper into the oral mucosal regions [5,11]. This is concurrent with our principle finding of ORN and MRONJ that they were further complicated by actinomycosis infection under conditions that led to deeper tissue access by this microorganism. 

Actinomycosis has been linked to the disease of osteonecrosis of the jaw in several studies [3,6,12,13,14,15,16]. The mainstay of actinomycosis management is prolonged antimicrobial treatment. High doses of antimicrobial administration are required since the sulphur granules colony of actinomycosis are thick and serves as a protective barrier. Hence, the necrotic and fibrotic surroundings of such infections require a larger diffusion gradient. Treatment should be vigorously combined with surgical removal of foci of infection and resection of bone sequestrum to expose healthy tissue. The drug of choice is penicillin and clindamycin with duration ranging from 3 to 12 months [17]. Another suggested regime is a prolonged treatment of high doses of penicillin G or amoxicillin for six to 12 months [11]. Consistently, our findings showed all specimens were with *Actinomyces* sp. Various literature also supported this finding. We therefore, would like to propose that all of the ORN and MRONJ cases (unless the histopathology show absence of *Actinomyces*), follow the regime indicated for the actinomycosis infection, which includes intravenous Penicillin G of 18-24mil units for two to six weeks followed by oral amoxicillin 500-750mg three or four times a day for six to twelve months until the infection subsides. Alternative antibiotics are eryhthromycin, doxycillin, clindamycin, and others.

*Actinomyces* sp. is an anaerobic Gram-positive bacterium with a filamentous shape. There are over 40 species of *Actinomyces* described, and 26 species have been implicated in human clinical infections and the most prevalent species found in humans is *Actinomyces israelii* [1]. The “companion microbes” are important in actinomycosis pathogenesis where these co-pathogen bacteria help to initiate and develop infection by inhibiting host defences or reducing oxygen tension, thus facilitating the anaerobic environment. Within this anaerobic environment, the vascular supply is destroyed and replaced by irregular, granular tissue, or sulphur granules, which is evident microscopically and this is a very important criterion in the diagnosis. This synergistic role is important since actinomyces have low virulence and invasiveness due to the lack of hyaluronidase enzyme which is responsible for host tissue degradation [5]. The destruction of host tissue is beneficial for these “companion microbes”, in providing nutrients and Fe^2+^ ions for these co-pathogen bacteria. The presence of foreign body-associated non-vital tissues raises the possibility of invasion even further. Additionally, the role of biofilm in this chronic infection is significant in promoting recurrence and protecting against the diffusion of antibiotic molecules. Strong biofilm production was evident involving several *Actinomyces*
*species* like *A. israelii, A. naeslundii*, and *A. viscosus* [1,5].

MRONJ includes bisphosphonate or other anti-resorptive drugs which work by slowing down bone loss in a certain type of disease, besides the exhaustive list of chemotherapeutic drugs. Theoretically, the breach of the oral mucosa promotes the local release of the drugs into the surrounding epithelium. An example would be in cases of trauma to the alveolus during dental extraction. This in turn will inhibit epithelial regeneration and may be a pathogenic factor in the occurrence of MRONJ [18]. Moreover, there is a diminished level of keratinocyte growth factor (KGF) which is strongly associated with unhealed epithelium. KGF has been proven to be highly capable of inducing not only growth but also migration of gingival epithelial cells [18]. Meanwhile, ORN typically develops after a few months or years of radiotherapy. It affects the blood vessels of bone, causing endarteritis that leads to thrombi formation and blocks the vessels. The interruption of collagen formation, an increased presence of free radicals, and impairment of wound healing also occur which ultimately causes bone necrosis. ORN incidence increases with the increased dosage of radiotherapy [3].

Actinomycosis is difficult to diagnose based solely on the clinical and radiological features. It may resemble malignancy, tuberculosis, or nocardiosis, as it continuously scatters and gradually forms a cold abscess. The gold standard method for diagnosing ONJ is the microscopic examination and bacterial culture of the abscess. Histopathological examination showed the bluish sulphur granules with radiating filaments surrounded by mixed inflammatory infiltrates [4]. The culture of *Actinomyces* is useless when the antibiotics are given prior to sample collection due to the significantly reduced sensitivity of the culture. In addition, the handling of specimens is very difficult as there could be inhibition of the *Actinomyces* growth by concomitant and/or contaminant microorganisms, inadequate culture condition, or inadequate short-term incubation, which increases the failure rate of the culture. Matrix-assisted laser desorption ionization time-of-flight (MALDI-TOF) has been claimed to be a quicker and more accurate tool for *Actinomyces* identification in the future; however, species identification remains uncertain [3]. Therefore, histological examination is preferred and is the best option for actinomycosis identification [9]. *Actinomyces* sp. are easily detected in the necrotic bone of specimens because of their peculiar morphology. Choi et al. (2022) reported the presence of *Actinomyces* sp. in about 70% of the samples from patients with MRONJ using histological techniques. Non-MRONJ patients showed a relatively low detection rate of *Actinomyces* (<20%) from the bone specimens, strongly implying that *Actinomyces* sp. are associated with the pathogenesis of MRONJ and not just as a superficial contaminant [4].

The analysis of the histological criteria was not a routine practice for the diseases. However, some of them were more prevalent in one or other diseases. There was also variability between our observation and other studies [12,13,19] since our sample size was small. The presence of necrotic bone was consistent in ORN and MRONJ specimens. Characteristics such as the presence of empty osteocyte lacunae, absence of osteoblastic rimming, and empty Haversian and Volkmann canals represented the general picture of bone necrosis. Osteoclastic cells were reported to have increased in number in MRONJ as bisphosphonate may have influenced its presence. Furthermore, they were observed to be detached from the bone margin, with a ruffled border, much rounder cell shape, and hypernucleated giant cells [19]. Meanwhile, our cases of MRONJ did not show osteoclasts with hypernucleated giant cells but much irregular shapes with ruffled borders. Unlike the other study [12], ORN did not show a large amount of both reactive bone and osteoblastic rimming. Inflammation was less frequently observed in the ORN specimens compared to other diseases which were not coherent with other reports [12]. Compromised vascularization was expected to be observed in both ORN and MRONJ, given the etiological factors of both diseases. Microorganisms of the *genus Actinomyces* were found in all samples, revealing no significant differences among the diseases. Consistently, *Actinomyces* were detected on the trabecular bone surfaces in MRONJ cases [12]. However, for theORN specimen, the microorganism was found in the medullary spaces. There was inconsistency in the number of microorganism colonies in both diseases which was similar with other study findings [12,13,19]. In addition, De Ceulaer et al. [13] proposed evidence supporting the hypothesis that MRONJ may be considered a bisphosphonate-induced actinomycosis infection of the jaw due to the following factors: (i) the high prevalence of isolation of *Actinomyces* from MRONJ bone lesions; (ii) the pathological similarities between MRONJ and *Actinomyces* osteomyelitis; (iii) the high incidence of normal mucosal barrier disruption as a necessary trigger to develop MRONJ in bisphosphonate-exposed patients; (iv) the predilection of bisphosphonate-induced osteonecrosis for the bones of the jaws; and (v) the favourable response of MRONJ on treatment that is active on *Actinomyces*. Therefore, actinomycosis has a major clinical implication for the prevention and treatment of MRONJ. 

A larger sample size is required to draw a significant relationship between actinomycosis in MRONJ or ORN. AN example would be with regard to the histological comparison of the presence of necrotic bone, inflammation, vascularization, and microorganism in ORN and MRONJ. This histological finding can further be associated with the clinical staging of ORN and MRONJ, thus aiding in the understanding of possible mechanisms and clinical progress.

## 5. Conclusions

The *Actinomyces* sp. was evident in the ORN and MRONJ sequestrum specimens. Its presence indicates prolonged antibiotic therapy. 

## Figures and Tables

**Figure 1 antibiotics-11-01067-f001:**
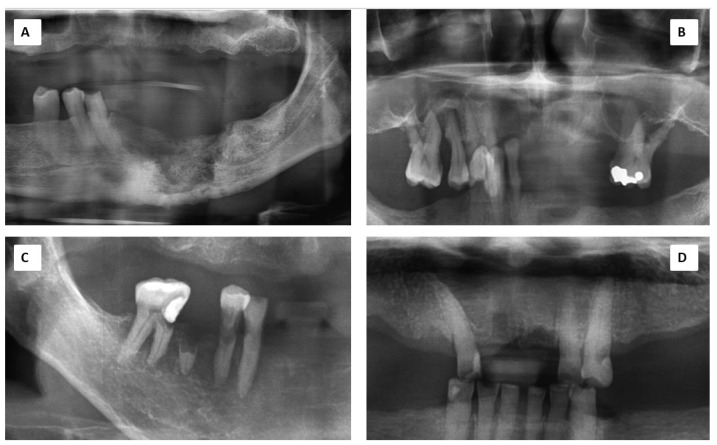
Orthopantomogram shows variety of presentations: (**A**): Case 1 showing osteolytic and sclerotic mandibular resorption; (**B**): Case 2 showing floating teeth indication severe maxillary bone resorption; (**C**): Case 6 showing severe bone resorption surrounding the 45, 46 and 47 teeth; (**D**): Case 3 showing normal bone trabeculae except at the extraction sockets on the anterior maxilla.

**Figure 2 antibiotics-11-01067-f002:**
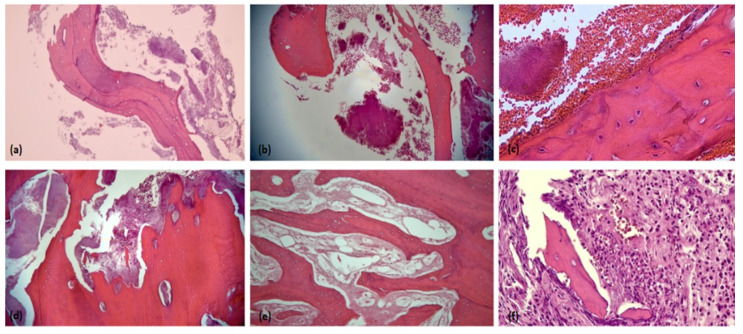
Photomicrographs H&E sections show (**a**) necrotic bone with basophilic reversal lines (100×); (**b**) areas of non-viable bone with clusters of microbial colonies (100×); (**c**) a higher magnification showing peripheral filamentous clubbing of the microorganism colonies (400×); (**d**) peripheral resorption of non-viable bone with abundance colonies of microorganism (100×); (**e**) non-viable bone with empty lacunae and avascular bone marrow (100×); (**f**) focal area of osteoclastic activity (400×).

**Figure 3 antibiotics-11-01067-f003:**
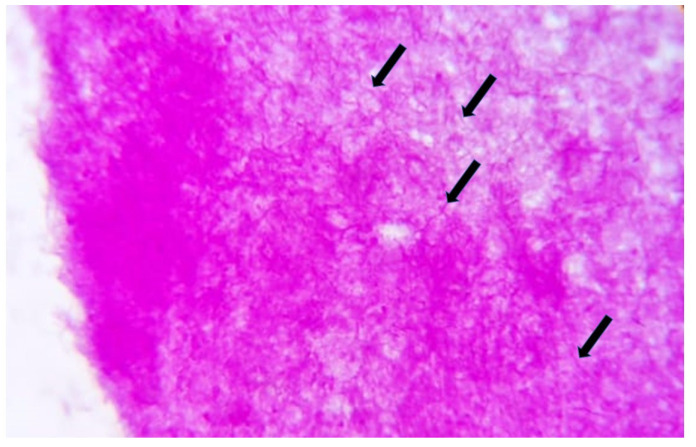
Periodic Acid Schiff (PAS) staining of the bacterial colonies is an adjunct special stain to further enhance the filamentous appearance of this anaerobic microorganism (black arrows) (PAS; objective magnification: 400×).

**Table 1 antibiotics-11-01067-t001:** Clinical and histological characteristics of the eight clinical cases of ORN and MRONJ.

Characteristics	Case 1	Case 2	Case 3	Case 4	Case 5	Case 6	Case 7	Case 8
Race	Malay	Malay	Chinese	Chinese	Malay	Chinese	Chinese	Chinese
Gender	Male	Male	Female	Female	Female	Male	Male	Male
Age	55	64	72	66	53	66	47	69
Comorbidities	GCT metastasis	Multiple myeloma, hypertension, dyslipidaemia	NPC, 8hypertension	Osteoporosis, SLE, diabetes, 8hypertension	NPC	Tongue OSCC	Lacrimal SCC	Tongue OSCC
Affected site	Mandible	Maxilla	Maxilla	Mandible	Mandible	Mandible	Maxilla	Mandible
Staging	Stage 2 *	Stage 3 *	Stage 3#	Stage 2 *	Stage 3#	Stage 1#	Stage 3#	Stage 3#
(i) Drugs involved in MRONJ, (ii) Month & year the drug started (iii) Month & year the drug completed	(i) Subcutaneous Denosumab monthly (ii) Sept 2019 (iii) Sept 2021	(i) Intravenous Zometa (ii) Feb 2015 (iii) Feb 2016 (i) Intravenous Denosumab monthly (ii) July 2020 (iii) Sept 2021		(i) Tablet Risedronate (ii) June 2007 (iii) Dec 2007 (i) Tablet Fosamax Plus (ii) Dec 2007 (iii) Oct 2011				
Radiotherapy doses			70 Gy		Data not available	66 Gy	Data not available	70 Gy
ORN onset post-radiotherapy			14 years		8 years	7 years	3 years	3 years
Radiographic findings	Mixed osteolytic and sclerosis at the lower left mandible edentulous alveolar ridge. Cortical bone margin intact.	Bone surrounding teeth 14-16 and 26 were severely resorbed. Alveolar ridge margin appeared shaggy with multiple radiolucencies.	No abnormality detected on the maxilla and mandibular ridges.	Sequestrum-like radiopacity seen on left mandible with a focal region of tooth 34 appeared radiolucent.	Mixed osteolytic and sclerosis at the angle of mandible. Cortical bone margin still intact. Bone supporting around teeth 46-47 are sufficient.	Mixed osteolytic and sclerosis at the lower edentulous ridges.	Unknown	Lower edentulous ridge showed general horizontal resorption.
Organisms isolated from culture & sensitivity	Mixed growth	No culture & sensitivity performed	No culture & sensitivity performed	No culture & sensitivity performed	Data not available	No culture & sensitivity performed	Data not available	No culture & sensitivity performed
Brief microscopic findings	Areas of acute inflammatory infiltrates with granulation tissue formation. Necrotic bone with sulphur granule-like material observed.	Necrotic bone with collection of bluish material resembling sulfur granules of actinomyces.	Fragments of non-vital mature bone surrounded by a focus of basophilic microorganisms with a mixed inflammatory infiltrate. Scalloping resorbed bone margin was also seen.	Presence of trabeculae of non-vital bones surrounded by bacterial colonies and mixed inflammatory cells was evident.	Non-vital bony fragments were surrounded by colonies of microorganisms and mixed inflammatory cells. Also observed were foci of necrosis with some areas exhibiting ingrowth of epithelium towards the non-vital bones.	Fragments of non-vital bone with peripheral resorption and numerous basophilic bacterial colonization.	Non-vital bones were observed to be attached to the non-inflamed fibrous tissue and there was presence of microorganism colonies.	Non-vital bones with attached intensely inflamed fibrous tissue and presence of microorganism colonies were present.
Final diagnosis	MRONJ	MRONJ	ORN	MRONJ	ORN	ORN	ORN	ORN

Staging * AAOMS 2022 [6]; # Lyons classification (2014) [10]. Abbreviations: GCT: giant cell tumour; NPC: nasopharyngeal carcinoma; SLE: Systemic lupus erythematosus; OSCC: oral squamous cell carcinoma; MRONJ: medication related osteonecrosis of the jaw; ORN: osteoradionecrosis.

**Table 2 antibiotics-11-01067-t002:** Histological findings of the ORN and MRONJ.

Histological Findings	ORN	MRONJ
Necrotic bone	✓	✓
Osteoclast	✓	✓
Osteoblastic rimming		✓
Reactive bone	✓	✓
Empty osteocytes lacunae	✓	✓
Inflammation	✓	✓
Blood vessels	✓	
Hyperaemia and thrombosis	✓	✓
Microorganism (*Actinomyces* sp.)	✓	✓

**✓**: indicates that the feature is present.

## Data Availability

Not applicable.

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
