# Peer review of "Actinomyces* sp. Presence in the Bone Specimens of Patients with Osteonecrosis of the Jaw: The Histopathological Analysis and Clinical Implication"

_antibiotics, 2022, doi:10.3390/antibiotics11081067_

Round 1
Reviewer 1 Report
This paper needs very extensive English language editing.
Line 57. Table 1 should appear in the results section, not the methods section.
Line 76. Table 2 should appear in the results section, not the methods section.
Please explain in the methods section how the Actinomycosis was detected.
Case 2. What medication was he on that led to the diagnosis of medication-related ONJ?
Lines 197-201. These sentences seem to have come from the article Ghaemzadeh, E., Aghahosseini, F., Mahdavi, N. N., & Razzazian, M. (2022). A Actinomycotic Osteomyelitis of the Maxilla in a Patient on Phenytoin. Frontiers in Dentistry. You need to cite it properly.
Line 202 " for all members of the groups" – It's not clear what groups you are referring to.
Line 289. "Our results showed all specimens were with Actinomyces sp." This should be at the start of the results section.
The conclusion is long and not well organized. I recommend following these guidelines from Clinical Microbiology and Infection (https://www.elsevier.com/journals/clinical-microbiology-and-infection/1198-743X/guide-for-authors): "We urge authors to structure their Discussion according to the recommendations of Docherty and Smith: BMJ 1999;318:1224-5; namely: summary of the principal findings; findings of the present study in light of what was published before; strengths and limitations of the study; meaning of the study; understanding possible mechanism; implications for practice or policy; implications for future research."
Reviewer 2 Report
The article addresses the topic of bone necroses, both from drugs and ionizing radiation, and their association with infection by Actinomyces species.
There are some important points to be taken into consideration:
1. The abstract does not reflect the content of the article.
2. There is no information about the ethical aspects and no submission to an ethics committee or authorization from the participants of the case reports.
3. For the diagnosis of osteonecroses there is AAOMS classification for MRONJ and there are classifications for osteoradionecrosis, which I suggest be used in the clinical classification and staging of the lesions of the cases presented.
4. It is very important for the drug-related case report to have information about the time of use of the drugs that trigger osteonecrosis. I suggest putting this information, as well as the post-radiotherapy time and the dose of radiation used in the jaws.
5. In Case 2 and some others, there is no clinical and imaging description of the clinical condition that led to the diagnosis of bone necrosis.
6. It is not clear in several clinical case descriptions about the conclusive diagnosis of the relationship of Actinomyces to bone necrosis, I suggest making this information clear.
7. Information about bacterial species or bacterial colonies in the case description is not compatible with the diagnosis Actinomycosis, As described in table 2.
8. Reference 14 has been updated by AAOMS in 2022, I suggest updating the reference, and reviewing the changes to put this information in the main text.
